# Highly selective detection of methanol over ethanol by a handheld gas sensor

J. van den Broek [1], S. Abegg [1], S.E. Pratsinis [1] & A.T. Güntner [1,2]

Methanol poisoning causes blindness, organ failure or even death when recognized too late. Currently, there is no methanol detector for quick diagnosis by breath analysis or for screening of laced beverages. Typically, chemical sensors cannot distinguish methanol from the much higher ethanol background. Here, we present an inexpensive and handheld sensor for highly selective methanol detection. It consists of a separation column (Tenax) separating methanol from interferants like ethanol, acetone or hydrogen, as in gas chromatography, and a chemoresistive gas sensor (Pd-doped $SnO_2$ nanoparticles) to quantify the methanol concentration. This way, methanol is measured within 2 min from 1 to 1000 ppm without interference of much higher ethanol levels (up to 62,000 ppm). As a proof-of-concept, we reliably measure methanol concentrations in spiked breath samples and liquor. This could enable the realization of highly selective sensors in emerging applications such as breath analysis or air quality monitoring.

---

[1] Particle Technology Laboratory, Department of Mechanical and Process Engineering, ETH Zurich, 8092 Zurich, Switzerland. [2] Department of Endocrinology, Diabetes, and Clinical Nutrition, University Hospital Zurich, 8091 Zurich, Switzerland. Correspondence and requests for materials should be addressed to A.T.G. (email: andreas.guentner@ptl.mavt.ethz.ch)

Ingestion, inhalation, or skin absorption of methanol leads to irreversible tissue damage, especially to eyes and nervous system, or even death[1]. This is attributed to metabolization of methanol to toxic formic acid and formaldehyde[2], if not immediately treated[3]. Especially in developing countries, methanol poisoning outbreaks occur frequently with hundreds of victims due to adulterated alcohol as shown recently in Iran (Oct. 2018, 959 cases)[4], Cambodia (May 2018, 237 cases)[5], and India (Feb. 2019, > 95 cases)[6]. Furthermore, methanol is often used as solvent or chemical feedstock[7] in laboratories and chemical plants, posing a potential hazard of intoxication.

The gold standard for detection of methanol intoxication is blood analysis by gas–liquid chromatography, but more frequent in hospitals is the indirect diagnosis through blood gas analysis[8]. However, both require trained personnel, are expensive and rarely available in developing countries where most outbreaks occur[9]. Blood methanol levels can also be determined non-invasively in exhaled breath[10], analogous to ethanol as widely applied by law enforcement[11]. The challenge is thereby the selective detection of methanol in the presence of much higher ethanol background typically present after consumption of tainted alcoholic beverages and during therapy where ethanol is used as an antidote[12]. Even more interesting might be simple methods for screening of alcoholic beverages to prevent methanol poisoning. But here too, the same challenge is met. Thus, inexpensive and portable devices are needed for rapid screening of methanol poisoning in breath and liquor by paramedics or even laymen.

Chemical gas sensors are promising due to their low cost[13], high miniaturization potential[14], and simple use[15]. In particular, metal-oxide sensors show high sensitivity when nanostructured, capable to detect analytes down to 5 ppb within seconds[16]. But, such sensors are typically non-selective[17], especially for chemically similar molecules (like methanol and ethanol), representing a long-standing challenge in the field. Therefore, current chemoresistive[18] and electrochemical[19] methanol gas sensors show cross-interferences to ethanol and other alcohols, hindering them for the targeted applications.

Filters can drastically improve the selectivity of chemical sensors by exploiting additional molecular properties of the target analyte. For instance, highly selective (>100–1,000) formaldehyde detection was possible even with a non-specific SnO$_2$-based sensor by placing ahead a microporous zeolite membrane to filter molecules by size[20]. This way, formaldehyde was detected down to 30 ppb in 90% relative humidity (RH) without interference of 1 ppm ammonia, acetone, isoprene, and ethanol[20]. Also a sorption packed bed separation column of polar, nanostructured alumina enabled separation of hydrophilic from hydrophobic compounds[21], analogous to a gas chromatographic (GC) column[22]. This has led to highly selective (>100) sensing of isoprene down to 5 ppb at 90% RH despite the presence of much higher (4–8 times) methanol, ammonia and acetone concentrations[21]. Non-polar adsorbents, such as Tenax TA, on the other hand, can separate molecules by their molecular weight and chemical functional groups[23]. They are widely used in air sampling, whereby heavy molecules are retained longer than lighter ones due to stronger adsorption by van-der-Waals forces[23]. Thus, they are also promising to separate methanol from ethanol as done already in GC for the analysis of liquor (e.g., detected by olfactometry with humans[24]) and human breath (e.g., by mass spectrometry[25]).

Here, we present a handheld and inexpensive methanol detector (Fig. 1a) capable to quantify methanol selectively in the presence of ethanol and other analytes (e.g., acetone, H$_2$). It consists of a small packed bed of Tenax (Fig. 1b) to separate the analytes and a highly sensitive, but non-specific microsensor (Fig. 1d) consisting of flame-made Pd-doped SnO$_2$ nanoparticles

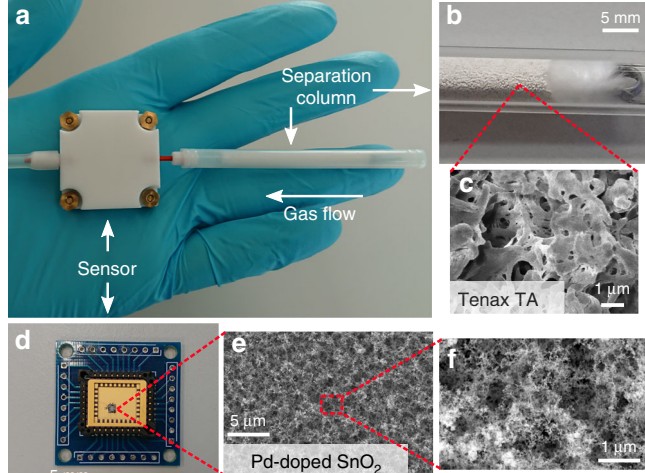

**Fig. 1** Images of the handheld methanol detector. **a** It consists of a microsensor (in Teflon housing) connected to a separation column (Tenax TA particles in Teflon tube). **b** Close-up of the separation column with particles inside a glass tube for better visibility. **c** Magnified images of a particle's surface. **d** The sensor chip carrier with a mounted microsensor. **e, f** Top-view images of the sensing films consisting of a fine network of agglomerated and aggregated Pd-doped SnO$_2$ nanoparticles

on interdigitated sensing electrodes. In comparison to typical GC instruments[22], our device is much smaller and less expensive. It is benchmarked by detection of methanol in the relevant concentration range in the presence of much higher ethanol levels at high RH. Ultimately, the methanol detector is tested to sense toxic methanol levels in tainted rum and even in spiked human breath.

## Results

**Detector design.** Figure 1a shows the handheld methanol detector. It consists of a separation column upstream of a micromachined metal-oxide gas sensor housed inside a Teflon chamber. Breath or the headspace of a beverage can be drawn by a pump through the separation column to the sensor. The separation column is a miniaturized GC column with Tenax TA as the stationary phase (shown in Fig. 1b) featuring lower adsorption strength to methanol over ethanol[23]. Figure 1c shows a scanning electron microscopy (SEM) image of a Tenax particle's surface revealing its high specific surface area (35 m$^2$ g$^{-1}$) and porosity (average pore size 200 nm). Compared to typical GC columns[22], the separation column used here is much shorter (4.5 cm) and thicker (4 mm inner diameter). Together with the small amount of adsorbent used (150 mg) and its large particle size (~200 μm), this results in a sufficiently small pressure drop (<20 mbar) to provide the required flow rate (25 mL min$^{-1}$) to the sensor.

Figure 1d shows the sensor bonded on a chip carrier. It is micromachined, offering small size and minimal power requirement (76 mW at 350 °C) readily suitable for integration into a handheld device. Figure 1e, f show top-view SEM images of the sensing film made of chemoresistive Pd-doped SnO$_2$ nanoparticles[26] offering high porosity and specific surface area (~80 m$^2$ g$^{-1}$ for similarly prepared Pt-doped SnO$_2$)[27]. The open film structure enables fast diffusion of analytes and interaction with the large surface area, important for rapid and highly sensitive methanol sensing. Such sensors have been used, for instance, for detection of only 3 ppb formaldehyde with fast response (140 s) and recovery (190 s) times, and good reproducibility (<10% response variation)[28].

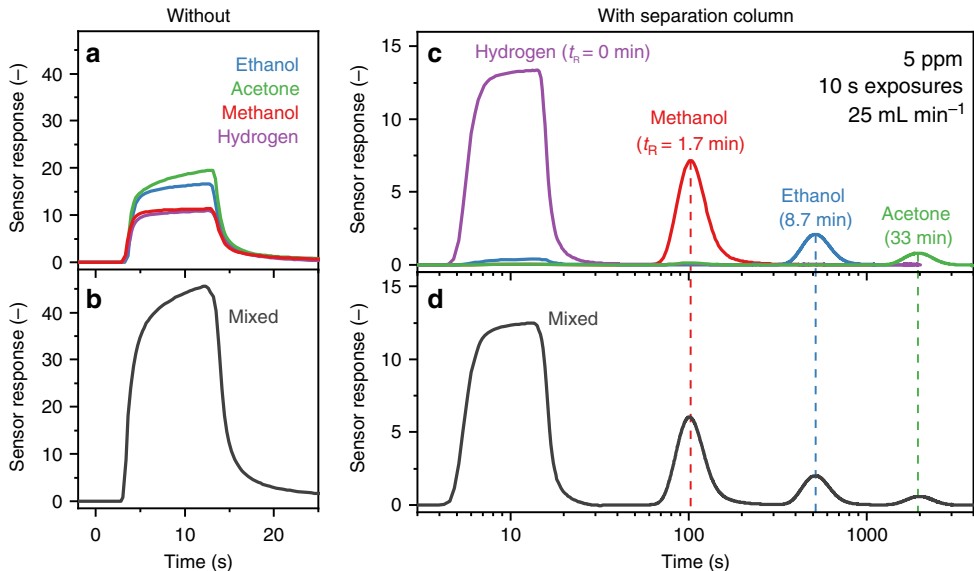

**Fig. 2** Single and mixed analyte responses without and with separation column. Response of the methanol detector **a**, **b** without and **c**, **d** with separation column to 10 s exposures of 5 ppm hydrogen (purple line), methanol (red line), ethanol (blue line), and acetone (green line) as well as their mixture (black line in **b** and **d**) at 50% RH and 25 mL min$^{-1}$ total flow rate. Also given are the corresponding retention times ($t_R$, dashed lines, for hydrogen $t_R = 0$ min as it is not retained). Note the different ordinate scale between **a**, **b** and **c**, **d**. The presence of the polymer sorbent packed bed upstream of the Pd-doped SnO$_2$ sensor facilitates the separation of the mixture components

**Selective methanol detection**. Figure 2a shows responses of the Pd-doped SnO$_2$ sensor without separation column to 10 s exposures of 5 ppm hydrogen (purple line), methanol (red line), acetone (green line), and ethanol (blue line) at 50% RH. The sensor quickly reacts to all these analytes with responses between 10 and 25. However, it cannot differentiate between them. This becomes even more evident when exposing the sensor to a mixture of these analytes (Fig. 2b). The sensor gives now a much higher response, slightly lower than the sum of the individual ones, as typically observed for chemoresistive sensors at such ppm concentrations[29]. As a consequence, this sensor cannot measure methanol selectively in the presence of such interferents.

When combined with the separation column, the sensor responses are separated as in a chromatograph. The response to hydrogen (purple line) remains the same (Fig. 2c). This is expected as hydrogen features low molecular weight and is not retained by Tenax[30]. For the other analytes, however, a different behavior is observed. In fact, methanol (red line) is now detected after >1 min with a maximum sensor response (i.e., retention time $t_R$, dashed lines) after 1.7 min. Note that the maximum methanol response is lower than without separation column, as the column dissipates it over a longer time period, in line with theory[31]. Most importantly, ethanol (blue line, $t_R = 8.7$ min) and acetone (green line, $t_R = 33$ min) are retained for much longer, in agreement with literature ($t_R = 2.2$, 10.8, and 36 min for methanol, ethanol, and acetone, respectively, at 20 °C)[30]. As a result, the separation column enables selective methanol detection. Interestingly also, the ethanol and acetone responses decrease with increasing $t_R$. In fact, the response to 5 ppm acetone is barely picked up by the sensor (while it was twice that of methanol without separation column, compare Fig. 2a, c).

When exposing the sensor with separation column to a mixture of the same analytes and concentrations (Fig. 2d), the analytes can be detected individually at their specific retention time with very high selectivity, identical to the single analyte exposures (Fig. 2c). Most remarkably, for the targeted applications, methanol is detected without ethanol interference, superior to state-of-the-art methanol sensors where the highest selectivity

to ethanol (>30) has been reported for imprinted Ag-doped LaFeO$_3$ core-shell particles[32].

As shown in Fig. 2c, d, the detector fully regenerates from each analyte or mixture exposure by flushing with air. The recovery time depends on the analyte and is about two to three times its retention time, in agreement with literature[33]. The recovery time can be decreased considerably by simply increasing the flow rate or by slight heating of the separation column (e.g., acetone from ~60 min to 20 s[30] when increasing column temperature and flow rate briefly to 80 °C and 100 mL min$^{-1}$)[23].

**Dynamic range**. Methanol concentrations in the targeted applications may occur from several ppm in breath[10] up to several hundred ppm in the headspace of beverages[34]. Figure 3 shows the Pd-doped SnO$_2$ sensor response with separation column when exposed for 10 s to 1–918 ppm of methanol at 50% RH. The median methanol concentration in healthy breath[35] (green line), and the range of exogenous[36] (orange line) and toxic breath methanol concentrations[1] (red area) are also indicated. The response curve is non-linear, in line with diffusion-reaction theory[29] for such semiconductive metal-oxide films at high analyte concentrations. Nevertheless, as a result, this separation column–sensor system can discriminate clearly toxic from non-toxic levels and even detect low concentrations of 1 ppm with a signal-to-noise ratio >100. Lower concentrations are not relevant for the liquor headspace and breath analyses, but such Pd-doped SnO$_2$ gas sensors can detect volatile organic compounds down to single ppb levels (e.g., 3 ppb formaldehyde[28]). In contrast, other benchtop methanol detectors (e.g., PTR-TOF-MS) can detect such low concentrations as well, but they have a much smaller dynamic range and require dilution to measure the high ppm concentrations present in breath or the headspace of beverages.

Please note that the response curve in Fig. 3 is valid for a separation column temperature of 22 °C at 50% RH. With increasing column temperature, the sensor responses become higher as $t_R$ decreases (Supplementary Fig. 1a). Most importantly, however, methanol is clearly separated and detected individually

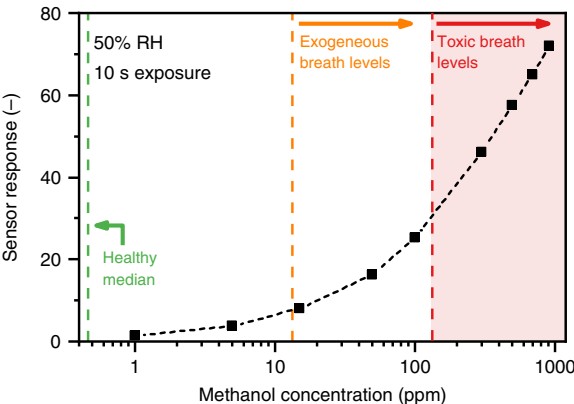

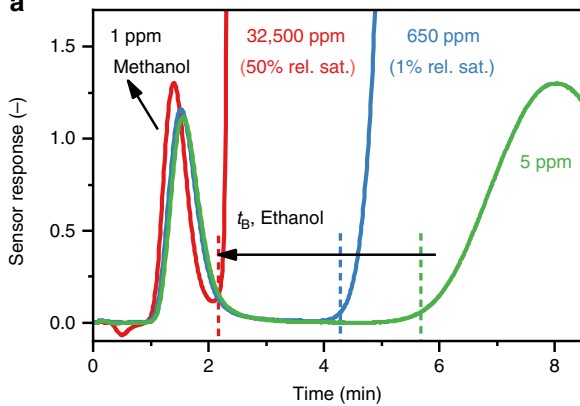

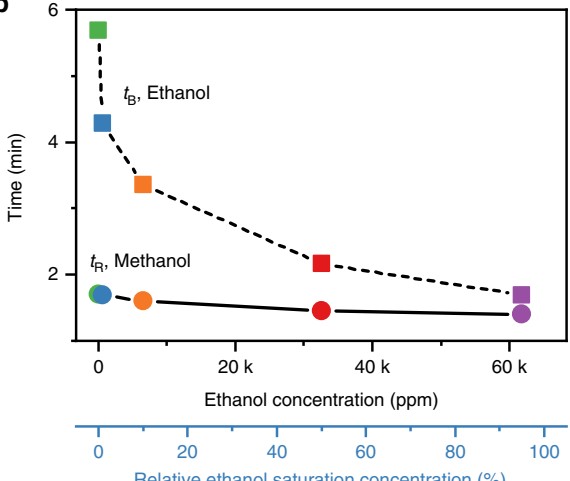

**Fig. 3** Dynamic range of the detector. Response of the detector (Tenax separation column + Pd-doped $SnO_2$ sensor) to 1–918 ppm methanol concentrations (black squares and dashed line). The median methanol concentration in healthy breath[35] (green dashed line), exogenous[36] (orange dashed line), and toxic[1] breath levels (red dashed line and shaded area) are indicated. Measurements were performed with 10 s exposure of all methanol concentrations at 50% RH and a flow rate of 25 mL min$^{-1}$ through the detector

from ethanol even at 40 °C. Such temperature effects could be accounted for by a temperature sensor. At higher humidity, $t_R$ does not change, in line with literature[23], but the responses decrease (Supplementary Fig. 1b). This is typical for such doped $SnO_2$ sensors[20] and can be addressed by using a sensor material less sensitive to humidity (e.g., Sb-doped $SnO_2$[37]) or by correction with a humidity sensor as done with sensor arrays to monitor volatile emission from human breath and skin[38].

**High ethanol background**. To analyze methanol in the headspace of alcoholic beverages or in intoxicated breath, the detector must remain accurate in the presence of very high ethanol concentrations. Figure 4a shows the response of the detector when exposed to 1 ppm methanol with interfering ethanol concentrations of 5 (green line), 650 (1% relative saturation, blue line), and 32,500 ppm (50% relative saturation, red line). Despite the significantly higher ethanol concentration, methanol is detected first ($t_R = 1.5$–1.7 min) giving comparable responses to the single gas calibration (Fig. 3). Ethanol is detected later with breakthrough times ($t_B$, dashed lines) that decrease with increasing concentration (5.7 min at 5 ppm to 2.2 min at 50% saturation) but are always higher than the $t_R$ of methanol. In GC, the same phenomenon is observed when overloading the column with analyte[39].

Interestingly, at 50% ethanol saturation concentration, methanol is detected slightly earlier with higher peak maximum and narrower peak width. Probably, this is due to competitive adsorption on Tenax where methanol is displaced by ethanol that adsorbs more strongly[40]. Nevertheless, the resulting error of 17% is sufficiently small for the targeted applications as the difference between normal and toxic methanol concentrations in liquor and breath are much larger (e.g., human breath median 0.46 ppm[35] vs. intoxicated >133 ppm[1]). If higher accuracy is required, alternatively, the area below the methanol response could be evaluated, as commonly done in gas chromatography[22]. In fact, the peak areas below the methanol responses are basically identical (within 2%), irrespective of the ethanol concentration (Supplementary Fig. 2).

Most importantly, the methanol response is clearly separated from that of ethanol even at very high concentrations. This is shown in Fig. 4b where the $t_R$ of methanol (solid line) and $t_B$ of

**Fig. 4** Methanol detection with high ethanol interference. **a** Responses of the methanol detector upon exposure to 1 ppm methanol in the presence of 5 (green line), 650 (1% relative saturation, blue line), and 32,500 ppm (50% relative saturation, red line) ethanol. Corresponding ethanol breakthrough times ($t_B$, dashed lines) are indicated. **b** The ethanol $t_B$ (squares and dashed line) and methanol retention time ($t_R$, circles and solid line) as a function of interfering ethanol concentration of 5 (green), 650 (1% relative saturation, blue), 6500 (10% relative saturation, orange), 32,500 ppm (50% relative saturation, red), and 62,000 ppm (95% relative saturation, purple)

ethanol (dashed line) are plotted for ethanol concentrations in the range of 5–62,000 ppm (95% saturation). The $t_B$ decreases exponentially with increasing concentration, in line with literature at lower concentrations[31]. Even at the most extreme conditions of 95% saturated ethanol atmosphere, methanol is detected independently of ethanol as its response is clearly separated from the breakthrough of ethanol. These results are astonishing considering the simplicity of this device and outperform other methanol detectors.

**Methanol-spiked liquor and breath**. Drinking as little as 6 mL methanol can be fatal[41]. Thus, a methanol detector for screening of alcoholic beverages could help to prevent methanol poisoning outbreaks. The safety threshold for naturally occurring methanol in liquor (40 vol% ethanol) is 0.4 vol% (US[34] and EU[36]), as such low levels are a byproduct of fermentation[34]. The detector must therefore be able to distinguish "safe" alcoholic beverages from tainted ones with typically much higher methanol content. Figure 5a shows the responses to pure (green line) and laced Arrack (common liquor in Southeast Asia) with 0.3 (blue line), 0.4

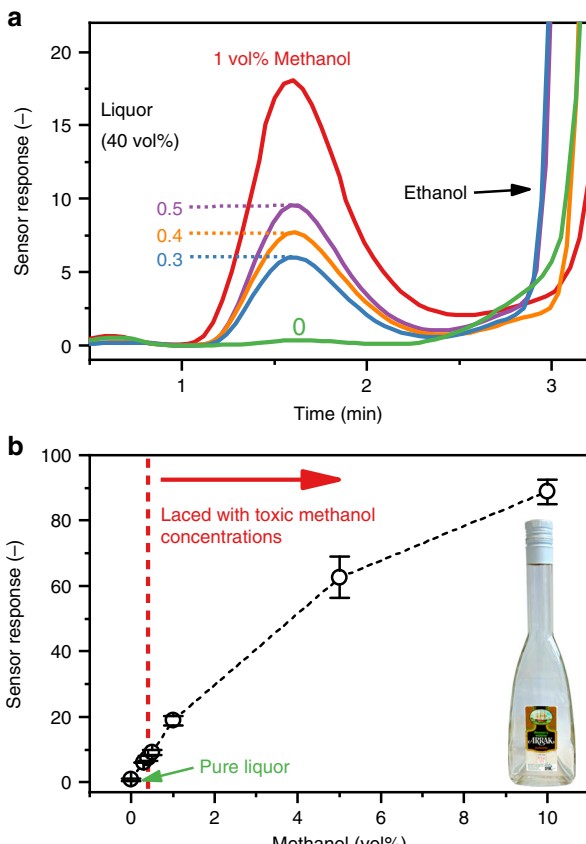

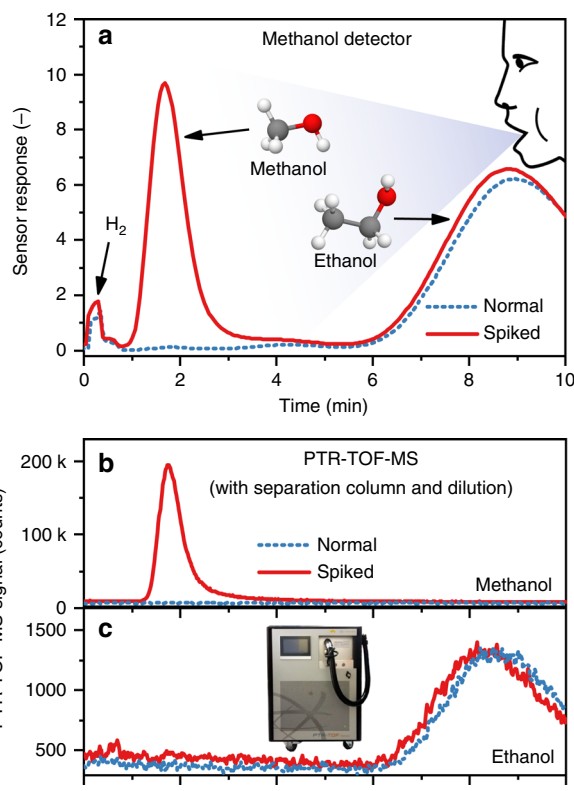

**Fig. 5** Detection of methanol in laced liquor (Arrack). **a** Response of the detector to pure liquor (green line) containing 40 vol% ethanol and laced with methanol of 0.3 (blue line), 0.4 (orange line), 0.5 (purple line), and 1 vol% (red line). **b** Sensor responses as a function of methanol content (0–10 vol%) in the liquor (black circles). The red dashed line indicates the legally allowed (US[34] and EU[36]) naturally occurring methanol content in liquor (40 vol% ethanol). Error bars indicate the standard deviation of at least three measurements (i.e., repeatability) with <15% variation

**Fig. 6** Methanol detection in spiked breath. **a** Response of the methanol detector to breath of an intoxicated volunteer (0.54‰ blood alcohol level) sampled from Tedlar bags. The blue dashed line shows the measurement from the normal breath sample and the red solid line from the spiked sample with 135 ppm methanol, indicating methanol intoxication. The PTR-TOF-MS measurements (with separation column and dilution) of the same samples for **b** methanol and **c** ethanol. The instrument shown in **c** is a PTR-TOF-MS 1000 (Ionicon, Austria) used for sensor validation. Hydrogen ($H_2$) is not retained by the separation column and does not interfere with the methanol detector

(orange line), 0.5 (purple line), and 1 vol% (red line) methanol. The detector clearly recognizes the added methanol at the expected $t_R = 1.7$ min, matching the retention time of methanol in laboratory gas mixtures (Figs. 2c and 4).

Most importantly, the response increases with increasing methanol concentration and even small differences between 0.3, 0.4 and 0.5 vol% (i.e., close to the allowed limit) can be clearly resolved by the sensor with high signal-to-noise ratio >100. In all cases, the response steeply increases after 3 min, corresponding to the high concentration of ethanol. Also at higher methanol contents of 5 and 10 vol% the sensor response continues to increase (Fig. 5b). As a result, the methanol detector can clearly distinguish pure Arrack from that laced with toxic levels of methanol with good repeatability (<15% variation, error bars in Fig. 5b). Owing to the high signal-to-noise ratio, also lower concentrations of methanol should be detectable, which may be interesting for the production monitoring and quality control of alcoholic beverages (e.g., naturally occurring methanol in wine[42]).

The methanol detector features also good stability with a sensor baseline drift of 0.7% per day during 18 days of testing (Supplementary Fig. 4). Such drifts could be corrected by an additional processing algorithm[43]. By purging with ambient air, it fully regenerates within 15 min (Supplementary Fig. 3a), enabling rapid screening and multiple uses with no observed saturation or

degradation effects over, at least, 2 weeks of repeated testing (Supplementary Fig. 4).

As a proof-of-concept for breath analysis, we evaluated the methanol detector on the original and methanol-spiked breath of an intoxicated (after ingestion of ethanol) volunteer (blood alcohol level 0.54‰ as measured with a Dräger Alcotest). Poisoning volunteers with methanol is unacceptable. However, spiking the analyte to the sample (i.e., standard addition method[44]) is a standard approach in analytical chemistry as the complexity of the gas matrix (i.e., intoxicated breath) is preserved. Figure 6a shows the detector response for the normal (blue dashed line) and methanol-spiked breath (135 ppm methanol, red solid line). Note that a methanol concentration of 135 ppm was chosen as it is just above the threshold of serious methanol intoxication (>133 ppm[1]). In both cases, the detector shows identical responses to hydrogen at $t = 0$–30 s (not retained) and ethanol ($t_R = 8.3$ min) with full recovery thereafter (Supplementary Fig. 3b). A clear peak associated with methanol is visible at $t_R = 1.7$ min with high signal-to-noise ratio (>1000), identical to laboratory gas mixtures (Fig. 2c). To verify the methanol (Fig. 6b) and ethanol (Fig. 6c) peaks, the same breath samples were analyzed by benchtop PTR-TOF-MS equipped with the same separation column. Note that these high concentrations were only measurable by PTR-TOF-MS by

additional dilution (please see Methods). Methanol and ethanol were detected at $t_R$ identical to the sensor, confirming the sensor results.

As a result, this detector can clearly differentiate between normal and methanol-spiked breath. Therefore, this it is promising for fast and non-invasive sensing of methanol poisoning. Given the high signal-to-noise ratio at 135 ppm methanol, it also shows promise for monitoring methanol elimination during treatment[10]. Of course, the results are rather preliminary (only one subject tested) and further validation with extended cohorts is required as done recently with breath acetone and a similar sensor (Si-doped $WO_3$) for body fat burn monitoring during exercise[45] and dieting[46].

Interestingly, in liquor (Fig. 5) and human breath (Fig. 6), only methanol, ethanol and hydrogen (breath) are clearly detected by the sensor, although both liquor[47] and breath[48] are complex mixtures with more than 100 and 800 analytes, respectively. This is probably due to the higher molecular weight and different functional groups (e.g., diols or glycols) of most interferants, resulting in longer retention in the separation column than methanol (e.g., ethylene glycol 100 times longer than methanol[30]). The most likely reason, however, is the much lower concentration of most confounders (e.g., 0.003 ppm trimethylamine in breath[49] compared to >133 ppm of methanol in case of intoxication[1]).

To the best of our knowledge, this is the first methanol sensor for the detection of relevant concentrations in the presence of ethanol in realistic samples of liquor and breath. Other sensors are either liquid sensors that cannot be used for breath (e.g., photoluminescent $Tb^{3+}$-based metal-organic framework sensor[50]), do not offer the required detection limit (e.g., Quartz tuning fork-based sensor[51]) or were not tested in gas mixtures (e.g., optical fiber sensor[52]).

## Discussion

We created an inexpensive, handheld and reliable methanol detector based on a separation column–sensor concept. The separation column is a small packed bed of polymer adsorbent (Tenax TA) that separates methanol from ethanol and other interferants including hydrogen and acetone analogous to a column in gas chromatography. So, methanol is detected within 2 min by a non-specific but highly sensitive nanostructured Pd-doped $SnO_2$ gas sensor in a wide concentration range from 1 to 918 ppm without interference of much higher ethanol concentrations (up to 62,000 ppm). The detector successfully quantified methanol concentrations in laced rum (Arrack) down to 0.3 vol% by analyzing its headspace and distinguished it from pure liquor. As first proof-of-concept, the detector was also tested on breath of an intoxicated volunteer, where it could clearly identify the sample spiked with toxic methanol concentrations. Thus, it shows promise for quick and non-invasive screening of methanol poisoning from breath and laced alcoholic beverages and could be used by first responders in developing countries, where most outbreaks occur.

In a broader sense, the present detector demonstrates how to possibly address a long-standing challenge of chemical sensors: the discrimination between analytes from the same chemical family. Giving comparable performance to a gas chromatographic column, such separation columns are much simpler in design, modular, and can be combined flexibly with other sensor technologies that often lack selectivity, such as optical sensors (e.g., plasmonic[53], fluorescent[54]), gas ionization detectors[55], electrochemical cells[56], and carbon-nanotube[57]- or graphene-based sensors[58]. Based on their small size and low price, such separation columns could enable highly selective, compact, and portable

gas detectors for emerging applications including medical breath analysis, food spoilage, and air quality monitoring.

## Methods

**Sensor fabrication.** Palladium-doped $SnO_2$ nanoparticles were produced by flame spray pyrolysis (FSP). So, Pd-acetylacetonate (Sigma-Aldrich, 99%) was dissolved in tin(-II-)ethylhexanoate (Strem Chemicals, ~90% in 2-ethylhexanoic acid) and xylene (Sigma-Aldrich, ≥98.5%) to obtain a total metal molarity (Pd and Sn) of 0.5 M and nominal Pd content of 1 mol%[28]. This precursor solution was fed through a capillary at 5 mL min$^{-1}$, dispersed into a fine spray by 5 L min$^{-1}$ oxygen (pressure drop of 1.6 bar) and ignited by a surrounding premixed methane/oxygen flame (1.25/3.2 L min$^{-1}$). The FSP reactor design is described in more detail elsewhere[26]. Nanoparticles were directly deposited[26] for 4 min onto micromachined free-standing membrane-type sensor substrates (1.9 × 1.7 mm², MSGS 5000i, Microsens SA, Switzerland) attached to a water-cooled holder at 20 cm height above the burner (HAB). The microsensor membranes feature an integrated heater layer underneath the interdigitated sensing electrodes. Subsequent in-situ annealing with a particle-free flame for 30 s at a HAB of 14.5 cm improved adhesion and cohesion of the highly porous sensing film[59]. Therefore, xylene was fed at 11 mL min$^{-1}$ through the nozzle with identical dispersion flow used during nanoparticle production. Finally, the sensors were annealed at 500 °C for 5 h in an oven (CWF13/23, Carbolite, United Kingdom) and wire-bonded onto leadless chip carriers (Chelsea Technology Inc., Massachusetts, US).

**Separation column fabrication.** The separation column is a packed bed of 150 mg Tenax TA (poly(2,6-diphenyl-*p*-phenylene oxide), 60–80 mesh, ~35 m² g$^{-1}$, Sigma Aldrich) packed inside a Teflon tube (4 mm inner diameter) and secured on both ends with silanized glass wool plugs and tension springs. Freshly prepared columns were flushed overnight with 100 mL min$^{-1}$ synthetic air (PanGas, $C_nH_m$ and $NO_x \leq 0.1$ ppm, Switzerland) at 50% RH to desorb impurities that might be adsorbed on the Tenax.

Scanning electron microscopy (SEM) images of the sensing film and the Tenax TA particle surface were made with a Hitachi S-4800 operated at 3 kV.

**Gas evaluation.** The methanol detector (Fig. 1a) consists of the separation column followed by the Pd-doped $SnO_2$ sensor. A miniature rotary vane pump of only 12 g (135 FZ 3 VDC Schwarz Precision, Germany) downstream of the sensor draws the sample through the separation column at 25 mL min$^{-1}$. The flow was validated by a calibrated bubble flow meter connected to the pump outlet. The sensor was heated by providing DC current (R&S HMC8043, Germany) through the heater of the micromachined sensor substrate. The sensing film temperature was set to 350 °C requiring only 76 mW. The ohmic resistance of the sensing film between the interdigitated electrodes was monitored with a multimeter (Keithley, 2700, USA). Sensor responses were evaluated as

$$S = \frac{R_A}{R_S} - 1 \qquad (1)$$

where $R_A$ and $R_S$ denote the sensor film resistances measured in background air (synthetic air or ambient air in case of breath and liquor headspace analysis) and during sample measurement, respectively. The retention time $t_R$ of an analyte was defined as the time from the start of analyte exposure to the sensor's maximum response, analogous to gas chromatography[60]. The breakthrough time $t_B$ of an analyte was defined as the time from the start of analyte exposure to an analyte response equal to 5% of the response to 1 ppm methanol.

For characterization of the sensor with synthetic gas mixtures, the methanol detector was connected to a gas delivery system illustrated schematically in Fig. 7. In specific, synthetic air was guided through a glass bubbler (Drechsel bottle, 125 mL, sintered glass frit, Sigma-Aldrich) containing ultrapure water (Milli-Q A10, Merck, Switzerland) and mixed with another stream of (dry) synthetic air to achieve 50% RH. All flows were accurately controlled by calibrated mass flow controllers (MFC, Bronkhorst, Netherlands) and the RH was verified by a humidity sensor (SHT2x, Sensirion AG, Switzerland). For generation of low analyte concentrations (5 ppm $H_2$, 1–5 ppm methanol, 5 ppm ethanol, and 5 ppm acetone), analytes were admixed from calibrated gas standards (PanGas, in synthetic air) and added to the synthetic gas stream through a septum via a capillary. Thereby, the capillary was quickly inserted into the septum for 10 s to generate well defined analyte exposures. The Teflon gas lines were heated to ~50 °C to avoid condensation and adsorption of water or analytes. The flow rates of the synthetic air and the analyte streams were varied in the range of 300–1000 and 1–300 mL min$^{-1}$, respectively, while the flow rate to the sensor was always kept constant by the pump at 25 mL min$^{-1}$.

For higher methanol concentrations (15–918 ppm), dry synthetic air was guided through a glass bubbler filled with ultrapure water and 1 vol% methanol (>99.9%, Sigma-Aldrich) and dilution with synthetic air. The generated methanol concentration from the bubbler was measured with a proton-transfer-reaction time-of-flight mass spectrometer (PTR-TOF-MS 1000, Ionicon, Austria) after further inlet dilution (1:200–1000) with synthetic air to avoid device saturation. The ionization conditions were 600 V drift voltage, 60 °C drift temperature, and 2.3 mbar drift pressure. Methanol concentrations were determined in the $H_3O^+$ mode

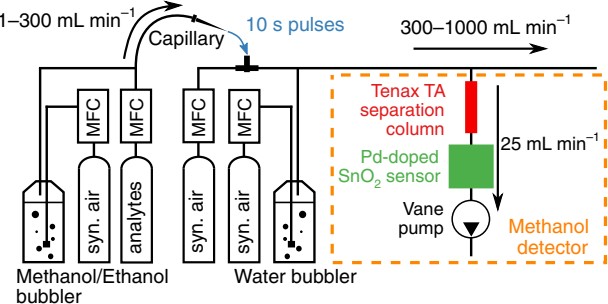

**Fig. 7** Schematic of the synthetic gas mixing setup and the methanol detector. The methanol detector (orange box) consisting of the packed bed separation column of polymer (Tenax TA, red) particles, followed by the chemoresistive (Pd-doped $SnO_2$, green) sensor and the vane pump that draws 25 mL min$^{-1}$ of gas sample. For characterization with synthetic gas mixtures, the detector is connected to a gas delivery system. It supplies the detector with a constant flow of humidified air by mixing dry and humidified synthetic air (syn. air). Analyte exposures are generated by admixing analytes from calibrated gas standards or by bubbling syn. air through analyte/water mixtures to the humidified syn. air stream with a capillary through a septum. Flows are accurately controlled by calibrated mass flow controllers (MFCs)

by measuring the counts per second at a mass-to-charge ratio[61] (m/z) of 33.0335 and comparison to a calibration curve obtained from the methanol gas standard. Higher ethanol concentrations (250–64,000 ppm) were generated similarly by bubbling air through pure ethanol (absolute, >99.8%, Fisher Chemical) and dilution with synthetic air. Generated concentrations were calculated from the weight loss of the bubbler after bubbling with air for 0, 2, 4, 6, and 8 h, while room temperature was kept constant at 22 °C.

**Evaluation of the headspace of drinks and human breath.** For testing of methanol-spiked drinks and breath, sensors were stabilized in ambient air with analyte background concentrations of methanol <50 ppb, ethanol <500 ppb, and acetone <100 ppb as determined by PTR-TOF-MS. Liquid samples were prepared in 25 mL glass bottles by mixing 5 mL of rum (40 vol% ethanol, Boven's echter Arrak, Indonesia) with 0, 0.3, 0.4, 0.5, 1, 5, and 10 vol% of methanol. Concentrations <0.3 vol% are not relevant for the liquor screening as the legal limit is 0.4 vol% in the US[34] and EU[36]. To guarantee equilibrium headspace concentrations, the samples were vigorously shaken manually for 30 s before sampling[62]. Headspace was sampled for 10 s by injecting a capillary attached to the methanol detector through a septum into the glass bottle caps. During sampling, a second capillary was inserted to keep the vial at ambient pressure.

For breath sampling, a volunteer consumed an alcoholic beverage (40 vol% ethanol, Bacardi Rum Carta Blanca) containing an equivalent of 50 mL pure ethanol. After 1 h, blood alcohol concentration was estimated with a breathalyzer (Alcotest 3820, Dräger, Germany). Another two breath samples were collected in Tedlar bags (3 L, SKC Inc., USA) by direct and complete exhalation through a Teflon tube. One of the bags was spiked with 300 mL of 918 ppm methanol in synthetic air (100% RH), giving a final concentration of 135 ppm, as verified by PTR-TOF-MS (inlet dilution 1:40 with synthetic air). Also to the second bag, 300 mL of synthetic air (100% RH) without methanol was added to keep dilution of the breath samples similar. Breath samples were stored no longer than 1 h in the Tedlar bags to avoid analyte losses[63]. The detector was exposed to breath samples for 10 s by injecting a capillary through a septum at the cap of the Tedlar bags. To validate the results of the methanol detector, the same breath samples were also analyzed by the PTR-TOF-MS (inlet dilution 1:40 with synthetic air) coupled to the separation column. Ethanol concentrations were determined in the $H_3O^+$ mode by measuring the counts per second at a mass-to-charge ratio (m/z) of 47.0490 (ref. [61]) and comparison to a calibration curve obtained from the ethanol gas standard. The experiment was approved by the ETH Zurich Ethics Commission and performed with written consent of the volunteer.

### Data availability

The data that support the findings of this study are available from the corresponding author upon reasonable request.

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

## Acknowledgements

Financial support by the Swiss National Science Foundation (Project Grants 159763 and 175754, and R'Equip Grant 170729) is gratefully acknowledged. We thank J. Winkler from the Optical Materials Engineering Laboratory (Prof. D. J. Norris, ETH Zurich) for help with the SEM. This research received the Best Poster Award in Exposure Measurement Methods and Techniques during the 2019 European Aerosol Conference, Aug. 25–30 in Gothenburg, Sweden.

## Author contributions

J.v.d.B. and A.T.G. conceived the concept and experiments. J.v.d.B. performed the experiments and the data evaluation. S.A. designed and provided the microsensors and contributed to the experimental design. S.E.P and A.T.G were in charge and advised on all parts of the project. J.v.d.B., S.A., S.E.P., and A.T.G co-wrote the paper. All authors gave final approval to the manuscript.

## Additional information

**Competing interests:** A patent application based on this manuscript has been submitted by J.v.d.B., S.A, S.E.P, and A.T.G.

**Peer Review Information** *Nature Communications* thanks Patrik Španěl and other, anonymous, reviewer(s) for their contribution to the peer review of this work. Peer reviewer reports are available.

