## [Peer Review File · Nature Communications]

Reviewers' comments:

Reviewer #1 (Remarks to the Author):

The manuscript describes a combination of a Tenax column filter with a non-specific sensitive sensor for selective analyses of methanol content in beverages and possibly also in exhaled breath. The details of the approach are to some degree original, however the advantage over previously reported techniques, addressing the construction of methanol/ethanol sensors, is not clearly demonstrated and some relevant work is not referenced. Overall, given the practical significance of this research and possible applicability of the presented technique to other applications, the paper may be suitable for publication after the authors clarify the originality over well established gas chromatography and address the following points by revising the manuscript.

1) Whilst the term filter is used to describe the concept, the function of the device is actually based on packed column gas chromatography. The key data look like chromatograms and the constant flow rate of 25 mL/min acts like mobile phase flow rate and the sorbent acts as the stationary phase. The description of this setup as a filter is thus not appropriate and the concept should be discussed in the terms of gas chromatography, a well-established technique. Appropriate references should be added.

2) Some previous work on methanol-ethanol sensors should be discussed, including

"Quartz tuning fork based portable sensor for vapour phase detection of methanol adulteration of ethanol by using aniline-doped polystyrene microwires" by Sampson et al. , *Microchimica Acta* 2017

"Photoluminescent Tb³⁺-based metal-organic framework as a sensor for detection of methanol in ethanol fuel" Fonseca et al. *J. Rare Earths* 2019

"High sensitivity optical fiber sensors for simultaneous measurement of methanol and ethanol" by Liu et al. *Sensors&Actuators B*, 2018

It is essential for publication in *Nature Communication* to demonstrate advantage over those previous studies on sensors.

3) The statement on line 72 that non-polar absorbents separate molecules by their molecular weight is not correct. There is some correlation with molecular weight, but other properties of the molecules are more important, consider ethanol and formic acid, surely their retention characteristics on Tenax will be different.

4) The statement on lines 165-167 is not relevant and should be deleted together with reference 34. PTR-MS is a trace gas analysis technique intended for sub ppbv analyses, it is obvious that 95% samples need to be diluted for its application. It would be more relevant to contrast the presented technique with the references mentioned in point 2) above.

5) Any research with human subjects needs independent ethics review by the appropriate committee prior to commencing the study. The subjects need to provide informed consent etc. The ethics aspects need to be disclosed and appropriate ethics committee approval must be quoted. There may be an exception when researchers experiment on themselves, but this would have to be clarified by an ethics expert.

6) Overall the excitement about separation of ethanol and methanol on a bed of sorbent through which air sample flows should be moderated, this is rediscovering gas chromatography after 70 years. Statements like "Most remarkably, methanol and ethanol were detected at tR identical to the sensor" are not justified, this is to be expected and very common.

7) How reliable and reproducible is the quantification? How sensitive is the response curve in Figure 3 to ambient temperature and sample humidity?

8) It may be interesting to quote the weight of the pump, 12 g only.

The originality of the manuscript is limited, as it combines well established GC principle with a non-specific sensor. However, the actual realisation of practically usable miniature device is interesting.

Patrik Spanel.

Reviewer #2 (Remarks to the Author):

This paper presents the combination of an absorbing (filter) medium and an electrochemical sensor for the selective detection of gas-phase methanol. One major novelty claimed is the development of a low-cost sensor that can distinguish between chemically similar species – a task that is intrinsically

difficult for gas sensors using various detection principles. Here, the gas sample is first transported through an adsorbing porous polymer medium that is used as a low-cost gas chromatograph and it is demonstrated that various species (especially hydrogen, acetone, methanol and ethanol) can be well separated based on their retention times on the minutes time scale. Therefore, the electrochemical sensor that would be unspecific for detection of the hydrocarbon compounds can selectively detect methanol even in the presence of a large excess of ethanol. The potential application field is the detection of methanol contamination in alcoholic drinks and methanol in breath analysis of intoxicated patients.

This concept of this “low-cost gas chromatograph” is strikingly simple but not new. The same authors have demonstrated an application for selective formaldehyde detection in Ref. 20. Therefore, the novelty of this paper is the described specific application for methanol. While this is an interesting field and the selection of absorber and detector (while not described in the paper) is certainly not straight forward – I doubt that the paper has the relevance required for publication in Nature Communications. This is also because the paper fails to clearly demonstrate the sensitivity that would be required for that specific application. I would recommend publication in a more specific sensor-related journal

Because the main content of the paper is not conceptual but more connected to the specific application, a number of questions arise:

- The “breath analysis” looks a bit made up. The “intoxicated person” is (for obvious reasons) not intoxicated with methanol but with ethanol only. The methanol detected in the breath analysis was added to the analyzed gas at a concentration of 135 ppm. Interestingly, figure 3 indicates that the toxic level is in the range of 13 ppm. Why is the demonstration then done at this exceedingly high concentration? The signal-to-noise in the measurements presented in Fig. 6 looks excellent – therefore, 135 ppm does not seem to represent the detection limit. However, what is the detection limit?
- For a sensor that is meant to decide between toxic or not, the accuracy is very important. How much does the methanol signal change in the presence of high concentrations of ethanol? (line 152 reports “comparable responses” in the presence of ethanol, referring to figure 3 (should be 4 I assume). This is insufficient, a quantitative analysis of the related error is necessary.
- The choice of the detector (Pt-doped SnO₂) is not discussed. What is its advantage compared to potential other (also commercial) materials or sensor concepts (also other than electrochemical)?
- Is there a strong temperature dependence of the retention times? (i.e., is temperature control required for the sensor)?
- If the filter is repeated for subsequent measurements: how long does purging take to prevent the release of unwanted compounds that might then overlap with the methanol signal?

(While for the medical application, a disposable filter could be possible, it does not seem reasonable to follow this path for a routine check for methanol-contaminated booze.)

- Is the strong deviation from linearity in Fig. 3 expected for this type of detectors? This nonlinearity would make the sensor susceptible to potential low-concentration contamination of the filter (see previous point)
- The sensitivity is also an open issue when looking at figure 5: The lowest methanol concentration measured is at 1 Vol.% Methanol, while the legal limit is at 0.4 Vol%. For the potential application described, wouldn't it be important to prove that a mixture is within the allowed limit? Therefore, what is the detection limit and if not yet reached, what are strategies to further reduce the limit?

Minor issues

Line 152: Figure 4 (not 3)

Line 184 normal

Line 190 were

Reviewer #3 (Remarks to the Author):

This paper is related to development of methanol detector with low cost metal oxide based gas sensor using chemoresistive gas sensor

(Pd-doped SnO₂ nanoparticles) to quantify the methanol concentration. Note that this paper used Tenax[®] Which is particularly useful for the analysis of high boiling compounds such as alcohols, polyethylene glycols, diols, phenols, monoamines and diamines, ethanolamines, aldehydes, ketones and chlorinated aromatics.

Tenax will trap other molecules with bigger molecular weight.

Therefore, this work need to show the specificity to Methanol over other molecule such as glycol , diols, amine etc. to strongly support the experimental results.

Another issue needs to address is the drift over time of sensors and absorbent to evident to quantitative performance of this sensor.

This paper will be suitable to more development or application based scheme journals because this paper do not present breakthrough research outputs that will suit for Nature Publication but if the novel material in sensor and absorbent are used , this paper will be able to publish.

Response to Referees of Manuscript NCOMMS-19-11951-T:

Response to Referee 1:

The manuscript describes a combination of a Tenax column filter with a non-specific sensitive sensor for selective analyses of methanol content in beverages and possibly also in exhaled breath. The details of the approach are to some degree original, however the advantage over previously reported techniques, addressing the construction of methanol/ethanol sensors, is not clearly demonstrated and some relevant work is not referenced. Overall, given the practical significance of this research and possible applicability of the presented technique to other applications, the paper may be suitable for publication after the authors clarify the originality over well established gas chromatography and address the following points by revising the manuscript.

1) Whilst the term filter is used to describe the concept, the function of the device is actually based on packed column gas chromatography. The key data look like chromatograms and the constant flow rate of 25 mL/min acts like mobile phase flow rate and the sorbent acts as the stationary phase. The description of this setup as a filter is thus not appropriate and the concept should be discussed in the terms of gas chromatography, a well-established technique. Appropriate references should be added.

Thank you for bringing this to our attention. It is true that our filter works like a packed bed in gas chromatography (GC), our apologies that this had not been stated explicitly. The term “filter” was used only because it is common in the community of gas sensors for components that remove or separate confounders. Nevertheless, to clarify this, we added now appropriate comparisons to GC throughout the manuscript. In specific:

In the Abstract: “... , as in gas chromatography, ...”, the keywords: “... Gas chromatography ...” and the Summary and Conclusions on p. 10, par. 4: “... analogous to a column in gas chromatography. ...” and on p. 11, par. 2: “... Giving comparable performance to a gas chromatographic column, such adsorption filters are much simpler in design, ...”

In the main text on p. 2, par. 2: “... analogous to a gas chromatographic (GC) column²² ...” and “... as done already in GC for the analysis of liquor (e.g. detected by olfactometry with humans²⁴) and human breath (e.g. by mass spectrometry²⁵). ...”, on p. 2, par. 3: “... In comparison to typical GC instruments,²² our device is much smaller and less expensive. ...”, on p. 3, par. 2: “...miniaturized packed bed GC column with Tenax TA as the stationary phase ...”, “Compared to typical GC columns,²² the filter is much shorter (4.5 cm) and thicker (4 mm inner diameter) ...”, on p. 4, par. 3: “... the sensor responses are separated as in a chromatograph. ...” and on p. 7, par. 1: “... In GC, the same phenomenon is observed when overloading the column with analyte.³⁹ ...”

2) Some previous work on methanol-ethanol sensors should be discussed, including ...

"Quartz tuning fork based portable sensor for vapour phase detection of methanol adulteration of ethanol by using aniline-doped polystyrene microwires" by Sampson et al. , Microchimica Acta 2017

This sensor was only tested in the headspace of ethanol/methanol lab mixtures with lowest methanol concentrations of 5 vol%. This is not sufficient for liquor testing where the allowed methanol concentration is typically 0.4 vol% (US³² and EU³⁴), as specified on p. 8, par. 1. Not to mention that it seems not capable to detect much lower methanol concentrations in human breath (e.g. >133 ppm indicating serious intoxication¹) as described on p. 6, par. 1.

In contrast, we show that our device differentiates between 0.3, 0.4 and 0.5 vol% of methanol in the presence of 40% ethanol with high signal-to-noise ratio (>100) in liquor (please see Figure R6 or new Figure 5 in manuscript).

"Photoluminescent Tb³⁺-based metal-organic framework as a sensor for detection of methanol in ethanol fuel" Fonseca et al. J. Rare Earths 2019

This is a liquid sensor that cannot operate in the gas phase. Sensing particles need to be admixed to the liquid and analyzed by a spectrofluorometer (not portable). As a result, this is not applicable for screening of methanol contamination in beverages or for intoxicated victims by breath analysis.

"High sensitivity optical fiber sensors for simultaneous measurement of methanol and ethanol" by Liu et al. Sensors&Actuators B, 2018

This sensor has only been tested for methanol and ethanol as single vapors. However, this is not the challenge as methanol has to be measured selectively in the presence of much higher ethanol concentrations, as stated explicitly on p. 1, par. 2.

... It is essential for publication in Nature Communication to demonstrate advantage over those previous studies on sensors.

As shown above, none of the listed methanol sensors has proven performance to detect realistic methanol/ethanol levels in liquor headspace (0.4 vol%³⁶ methanol at 40 vol% ethanol) and human breath (133 ppm¹ in drunken breath) samples.

Nevertheless, to address these sensor types and demonstrate the advantage of the presented methanol detector, we made the following addition to the manuscript on p. 10, par. 3: "... To the best of our knowledge, this is the first methanol sensor for the detection of relevant concentrations in the presence of ethanol in realistic samples of liquor and breath. Other sensors are either liquid sensors that cannot be used for breath (e.g. photoluminescent Tb³⁺-based metal-organic framework sensor⁵⁰), do not offer the required detection limit (e.g. Quartz tuning fork based sensor⁵¹) or were not tested in gas mixtures (e.g. optical fiber sensor⁵²). ..."

3) The statement on line 72 that non-polar absorbents separate molecules by their molecular weight is not correct. There is some correlation with molecular weight, but other proprieties of the molecules are more important, consider ethanol and formic acid, surely their retention characteristics on Tenax will be different.

We agree and added on p. 2, par. 2: "... and chemical functional groups.²³ ..."

4) The statement on lines 165-167 is not relevant and should be deleted together with reference 34. PTR-MS is a trace gas analysis technique intended for sub ppbv analyses, it is obvious that 95% samples need to be diluted for its application. It would be more relevant to contrast the presented technique with the references mentioned in point 2) above.

We agree and removed this passage on p. 7, par. 3.

5) Any research with human subjects needs independent ethics review by the appropriate committee prior to commencing the study. The subjects need to provide informed consent etc. The ethics aspects need to be disclosed and appropriate ethics committee approval must be quoted. There may be an exception when researchers experiment on themselves, but this would have to be clarified by an ethics expert.

According to the ETH Ethics Commission this project is not subject to approval (letter attached at the end of the document).

We also added on p. 15, par. 2: "... The experiment was approved by the ETH Zurich Ethics Commission and performed with written consent of the volunteer. ..."

6) Overall the excitement about separation of ethanol and methanol on a bed of sorbent through which air sample flows should be moderated, this is rediscovering gas chromatography after 70 years. Statements like "Most remarkably, methanol and ethanol were detected at tR identical to the sensor" are not justified, this is to be expected and very common.

We removed "Most remarkably" on two occasions on p. 9, par. 2.

7a) How reliable and reproducible is the quantification?

For our methanol detector, the intrasample repeatability of the measurement on Arrack liquor had less than 15% variation in the tested concentration range as shown by the error bars in Figure R1 (or new Figure 5b). This is more than sufficient for the application to differentiate pure (<0.4 vol% methanol) from laced liquors with toxic methanol concentrations (typically >1 vol%), as well as normal (median 460 ppb³⁵) from toxic breath methanol concentrations (>133 ppm¹). The reproducibility of the response to such VOCs of the Pd-SnO₂ gas sensors had less than 10% variation²⁸.

Figure R1 | Response of methanol detector to Arrack as a function of its methanol content. Error bars indicate the repeatability of at least three measurements with less than 15% variation.

In response, Figure 5b was replaced by Figure R1 and we added to the text on p. 3, par. 3: "... Such sensors had been used, for instance, for the detection of only 3 ppb formaldehyde with fast response (140 sec) and recovery (190 sec) times and good reproducibility (<10% response variation).²⁸ ..." and on p. 8, par. 2: "... with good repeatability (<15% variation, error bars in Figure 5b). ..."

7b) How sensitive is the response curve in Figure 3 to ambient temperature ...

We tested this now for our filter by external heating at 22, 30 and 40 °C (Figure R2). With higher temperature, elution of analytes is faster and peaks spread less, as expected³¹. This leads to a stronger (maximum) sensor response (e.g. 44% higher from 22 to 30 °C). However, even at 40 °C, the responses of methanol and ethanol are separated and detected individually.

To clarify this, we added Figure R2 as Figure S1a to the new Supporting Information and added to the text on p. 6, par. 2: "... Please note that the response curve in Figure 3 is valid for a filter temperature of 22 °C and at 50% RH. With increasing filter temperature, the sensor responses become higher as t_R decreases (Figure S1a). This can be corrected with an additional temperature sensor. Importantly, however, methanol is clearly separated and detected individually from ethanol even at 40 °C. Such temperature effects could be accounted for by a temperature sensor. ..."

Figure R2 | Response of the methanol detector to 10 sec pulses of 5 ppm methanol (solid lines) and ethanol (dashed lines) at 22 (blue), 30 (green) and 40 °C (red) column temperature.

7c) ... and sample humidity?

Relative humidity adsorbs only weakly on Tenax TA and thus has a negligible effect on retention times of polar analytes.³¹ We validated this now at 10, 50 and 90% RH (Figure R3) and confirm that retention times stay constant within the repeatability of the measurement. However, the methanol sensor responses decrease at higher humidity (e.g. by 40% from 10 to 90% RH). This is typical for such SnO₂-based sensors.²⁰ It can be addressed by using a sensor material with less cross-sensitivity to humidity (e.g. Sb-doped SnO₂³⁷) or by correction with a humidity sensor as done with sensor arrays when monitoring volatile emission from human breath and skin.³⁸

To clarify this, we added Figure R3 as Figure S1b to the Supporting Information and included in the text on p. 6, par. 2: "... At higher humidity, t_R does not change, in-line with literature,²³ but the responses decrease. This is typical for such SnO₂-doped sensors²⁰ and can be addressed by using a sensor material less sensitive to humidity (e.g. Sb-doped SnO₂³⁷) or by correction with a humidity sensor as done with sensor arrays to monitor volatile emission from human breath and skin.³⁸ ..."

Figure R3 | Response of detector to 10 sec pulses of 5 ppm methanol (solid lines) and ethanol (dashed lines) at 90 (blue), 50 (green) and 10 %RH (red) of the gas flow.

8) It may be interesting to quote the weight of the pump, 12 g only.

We added to p. 12, par. 4: "... of only 12 g ..."

The originality of the manuscript is limited, as it combines well established GC principle with a non-specific sensor. However, the actual realisation of practically usable miniature device is interesting.

Patrik Spanel.

Thank you for the valuable comments and hope we have addressed them convincingly.

Response to Referee 2:

This paper presents the combination of an absorbing (filter) medium and an electrochemical sensor for the selective detection of gas-phase methanol. One major novelty claimed is the development of a low-cost sensor that can distinguish between chemically similar species – a task that is intrinsically difficult for gas sensors using various detection principles. Here, the gas sample is first transported through an adsorbing porous polymer medium that is used as a low-cost gas chromatograph and it is demonstrated that various species (especially hydrogen, acetone, methanol and ethanol) can be well separated based on their retention times on the minutes time scale. Therefore, the electrochemical sensor that would be unspecific for detection of the hydrocarbon compounds can selectively detect methanol even in the presence of a large excess of ethanol. The potential application field is the detection of methanol contamination in alcoholic drinks and methanol in breath analysis of intoxicated patients.

1a) This concept of this "low-cost gas chromatograph" is strikingly simple but not new. ...

We regret to not clearly describe the novelty of our work. Indeed, applying a packed column to separate methanol from ethanol is analogous to well-established gas chromatography (GC). We now added a detailed comparison of our detector to GC throughout the manuscript (please refer to Referee 1, comment #1 for details).

1b) ... Therefore, the novelty of this paper is the described specific application for methanol. While this is an interesting field and the selection of absorber and detector (while not described in the paper) is certainly not straight forward – I doubt that the paper has the relevance required for publication in Nature Communications ... I would recommend publication in a more specific sensor-related journal.

The high practical significance and interdisciplinary impact of our work should suit Nature Communications. In fact, we might be the first to present a practical, hand-held and inexpensive methanol detector for liquor (Figure 5) and breath (Figure 6). This might be a breakthrough in analytical chemistry as previous sensors failed at this task (please see Referee 1, comment #2 for specific examples).

1c) ... The same authors have demonstrated an application for selective formaldehyde detection in Ref. 20 ...

Our formaldehyde sensor²⁰ does not apply a sorption packed bed (analogous to gas chromatography) but a molecular sieve membrane. Therefore, the separation principle is fundamentally different (separation by molecular size vs. sorption properties), as described on p. 2, par. 2.

1d) ... This is also because the paper fails to clearly demonstrate the sensitivity that would be required for that specific application ...

We added new data to demonstrate that our sensor has the required sensitivity for breath and liquor headspace analysis (please see our detailed responses to your comments #3, #5, #6 and #8).

Because the main content of the paper is not conceptual but more connected to the specific application, a number of question arise:

2a) The “breath analysis” looks a bit made up. The “intoxicated person” is (for obvious reasons) not intoxicated with methanol but with ethanol only.

Correct, as intoxicating a volunteer with methanol is unacceptable. However, to simulate this condition as close as possible, we asked a volunteer to drink 120 mL of rum (40 vol% ethanol). Relevant concentrations of methanol (135 ppm) indicating the need for immediate treatment¹ were then added to the sampled breath, as described (p. 14, par. 3). This is a standard approach in analytical chemistry (the so-called “standard addition method”⁴⁴) as the complexity of the gas mixture (drunken breath) is preserved.

To avoid possible misunderstandings, we added on p. 9, par. 1: “... Intoxicating volunteers with methanol is unacceptable. However, spiking the analyte to the sample (i.e. standard addition method⁴⁴) is a standard approach in analytical chemistry as the complexity of the gas matrix (i.e. drunken breath) is preserved. ...”

2b) The methanol detected in the breath analysis was added to the analyzed gas at a concentration of 135 ppm. Interestingly, figure 3 indicates that the toxic level is in the range of 13 ppm. Why is the demonstration than done at this exceedingly high concentration? The signal-to-noise in the measurements presented in Fig. 6 looks excellent – therefore, 135 ppm does not seem to represent the detection limit.

Thank you for bringing this to our attention. Actually, the 13 ppm indicated in Figure 3 are not typical breath levels for a methanol intoxication but rather describe exogenous³⁶ breath methanol levels. Serious

intoxication requiring immediate treatment is required above 133 ppm according to The American Academy of Clinical Toxicology.¹ As shown in Figure 6a and described on p. 9, par. 1, the methanol detector can clearly distinguish such toxic (i.e. 135 ppm, red line) from normal (blue line) levels. As pointed out correctly, the signal-to-noise ratio of this measurement is indeed very high (>1000), so also lower concentrations can be detected. This is attractive for monitoring methanol elimination during treatment.¹⁰

Accordingly, we modified Figure 3 indicating now the exogeneous and toxic breath level ranges separately. We also adjusted the text on p. 5, par. 3: "... exogeneous³⁶ (orange line) and ..." and on p. 9, par. 1: "... Note that a methanol concentration of 135 ppm was chosen as it is just above the threshold of serious intoxication (>133 ppm¹). ..." and "... with high signal-to-noise ratio (>1000), ...", and on p. 9, par. 2: "... Given the high signal-to-noise ratio at 135 ppm methanol, it also shows promise for monitoring methanol elimination during treatment.¹⁰ ..."

2c) However, what is the detection limit?

Here, we tested our filter-sensor system down to 1 ppm in synthetic gas mixtures (Figure 3), as lower concentrations are not relevant for the liquor headspace and breath analyses. However, such Pd-doped SnO₂ gas sensors can detect analytes down to single ppb levels as demonstrated, for instance, down to 3 ppb formaldehyde for indoor air monitoring.²⁸

This information was added on p. 5, par. 3: "... Lower concentrations are not relevant for the liquor headspace and breath analyses, but such Pd-doped SnO₂ gas sensors can detect volatile organic compounds down to single ppb levels (e.g., 3 ppb formaldehyde²⁸). ..."

3) For a sensor that is meant to decide between toxic or not, the accuracy is very important. How much does the methanol signal change in the presence of high concentrations of ethanol? (line 152 reports "comparable responses" in the presence of ethanol, referring to figure 3 (should be 4 I assume). This is insufficient, a quantitative analysis of the related error is necessary.

The response error to 1 ppm methanol is 17% when the interfering ethanol concentration is increased from 5 to 32,500 ppm (Figure 4a). This is accurate enough for the targeted applications as the difference between normal and toxic methanol concentrations in liquor and breath are typically much larger (breath median 0.46 ppm³⁵ vs. intoxicated >133 ppm¹). However, the peak areas below the methanol responses are the same (within 2%) irrespective of the ethanol concentration (Figure R4). Thus, alternatively, the area below the methanol response could be evaluated, as commonly done in gas chromatography.²²

In response, we added Figure R4 as Figure S2 to the Supporting Information. Also, we added to the text on p. 7, par. 1: "... Nevertheless, the resulting error of 17% is sufficiently small for the targeted applications as the difference between normal and toxic methanol concentrations in liquor and breath are much larger (e.g. human breath median 0.46 ppm³⁵ vs. intoxicated >133 ppm¹). If higher accuracy is required, alternatively, the area below the methanol response could be evaluated, as commonly done in gas chromatography.²² In fact, the peak areas below the methanol responses are basically identical (within 2%), irrespective of the ethanol concentration (Figure S2). ..."

Figure R4 | Area below the response curve of the methanol detector upon exposure to 1 ppm methanol in the presence of 5 (green), 650 (1% relative saturation, blue) and 32,500 ppm (50% relative saturation, red) ethanol. While the maximum peak height is slightly higher with high ethanol interference (see Figure 4a), the areas below the methanol response curves are equal when evaluated just before the breakthrough (t_B) of ethanol.

4) *The choice of the detector (Pt-doped SnO₂) is not discussed. What is its advantage compared to potential other (also commercial) materials or sensor concepts (also other than electrochemical)?*

Chemical gas sensors, in particular metal-oxide sensors, have low cost,¹³ are easy to miniaturize¹⁴ and simple in use,¹⁵ as described on p. 1, par. 3. We used a Pd-doped SnO₂ sensor as it features also high sensitivity to volatile organic compounds to detect even the lowest concentrations (e.g. 3 ppb formaldehyde) with fast response (140 sec) and recovery (190 sec) times and good reproducibility (<10% response variation).²⁸

We added this to p. 3, par. 3: "... Such sensors were used, for instance, for detection of only 3 ppb formaldehyde with fast response (140 sec) and recovery (190 sec) times and good reproducibility (<10% response variation).²⁸ ..."

5) *Is there a strong temperature dependence of the retention times? (i.e., is temperature control required for the sensor)?*

With increasing temperature, the retention times on Tenax TA become shorter.³¹ In response, we tested our filter now by external heating at 22, 30 and 40 °C (please see above Figure R2). With higher temperature, elution of analytes is faster and the peaks are spread less, leading to a higher maximum response. Most importantly, however, the responses of methanol and ethanol are separated and detected individually even at 40 °C. If this needs to be accounted for, an additional temperature sensor could be applied.

To clarify this, we added Figure R2 as Figure S1a to the Supporting Information and to the text on p. 6, par. 2: "... With increasing filter temperature, the sensor responses become higher as t_R decreases (Figure S1a). Most importantly, however, methanol is clearly separated and detected individually from ethanol even at 40 °C. Such temperature effects could be accounted for by a temperature sensor. ..."

6) If the filter is repeated for subsequent measurements: how long does purging take to prevent the release of unwanted compounds that might than overlap with the methanol signal? (While for the medical application, a disposable filter could be possible, it does not seem reasonable to follow this path for a routine check for methanol-contaminated booze.)

After a measurement, the sensor baseline is recovered within 15 min by just purging with ambient air, as shown in Figure R5a for methanol-laced Arrack and in Figure R5b for methanol spiked-breath. Then the filter can be re-used. This is short enough for liquor or breath screening. If necessary, regeneration time could be decreased by increasing the flow rate or slightly heating the filter as described on p. 5, par. 2. The filter is multi-use as we did not observe any saturation or degradation effects over, at least, two weeks of repeated testing.

Figure R5 | Response of the detector to (a) vapor from Arrack liquor containing 40 vol% ethanol and laced with 5 vol% methanol and (b) breath of a drunken volunteer spiked with 135 ppm methanol that is equivalent to the breath of a person being drunk with the above laced liquor according to the standard addition method⁴⁴. The 136 ppm ethanol correspond to a blood alcohol level of 0.54 ‰ according to the Dräger Alcotest 3820 device manual. The sensor baseline is recovered in both cases within 15 min by flushing with ambient air.

In response, we added Figure R5 as Figure S3 to the Supporting Information. We also added on p. 8, par. 3: “... By purging with ambient air, the detector fully regenerates within 15 min (Figure S3a), enabling rapid screening and multiple uses with no observed saturation or degradation effects over, at least, two weeks of repeated testing (Figure S4). ...” and on p. 9, par. 1: “... with full recovery thereafter (Figure S3a). ...”

7a) Is the strong deviation from linearity in Fig. 3 expected for this type of detectors?

Yes, this is expected. According to diffusion-reaction theory²⁹ for such semiconductive metal-oxide film sensors, response curves become non-linear at high analyte concentrations.

We added this to p. 5, par. 3: “... The response curve is non-linear, in-line with diffusion-reaction theory²⁹ for such semiconductive metal-oxide films at high analyte concentrations. ...”

7b) This nonlinearity would make the sensor susceptible to potential low-concentration contamination of the filter (see previous point).

The performed measurements on commercial Arrack liquor (Figure 5) and breath samples (Figure 6) do not show much of an impact of confounders on methanol detection even though exhaled breath⁴⁸ contains

more than 800 and the headspace of liquor⁴⁷ more than 100 analytes. However, they occur at much lower concentrations (e.g. 0.003 ppm trimethylamine in breath⁵⁰ compared to >133 ppm of methanol in case of intoxication¹) and most are retained longer and dispersed more by the filter.³¹ As a result, their response on the sensor becomes negligible as shown from our breath and liquor measurements.

In response, we added on p. 10, par. 1: "...Interestingly, in liquor (Figure 5) and human breath (Figure 6), only methanol, ethanol and hydrogen (breath) are clearly detected by the sensor, although both liquor and breath are complex mixtures with more than 100⁴⁷ and 800⁴⁸ analytes, respectively. This is probably due to the higher molecular weight and different functional groups (e.g. diols or glycols) of most interferants, resulting in longer retention in the filter than methanol (e.g. ethylene glycol 100 times longer than methanol³⁰). Most likely reason, however, is the much lower concentration of most confounders (e.g. 0.003 ppm trimethylamine in breath⁵⁰ compared to >133 ppm of methanol in case of intoxication¹). ..."

8a) The sensitivity is also an open issue when looking at figure 5: The lowest methanol concentration measured is at 1 Vol.% Methanol, while the legal limit is at 0.4 Vol%. For the potential application described, wouldn't it be important to prove that a mixture is within the allowed limit?

The detection limit is indeed much lower. Figure R6 shows additional measurements of commercial Arrack liquor laced with 0.3 (blue), 0.4 (orange), 0.5 (purple) and 1 vol% methanol (red). These low concentrations can be detected and clearly differentiated with high (>100) signal-to-noise ratio. As a result, the methanol detector can recognize Arrack samples with legal (<0.4 vol% from natural fermentation³⁴) and illegal amounts (>0.4 vol%, US³⁴ and EU³⁶) of methanol.

In response, we replaced Figure 5a with Figure R6 and added to the Methods on p. 14, par. 3: "...0.3, 0.4, 0.5 ...". We also modified the main text on p. 8, par. 1: "... 0.3 (blue), 0.4 (orange), 0.5 (purple) and 1 vol% (red) ...", "... and even small differences between 0.3 to 0.4 and 0.5 vol% (i.e. close to the allowed limit can be clearly resolved by the sensor with high signal-to-noise ratio >100. ...)" and "... Also at higher methanol contents of 5 and 10 vol% the sensor response continues to increase (Figure 5b). ...". Adjustments were also made to the Summary and Conclusions on p. 10, par. 3: "...0.3 vol% ..."

Figure R6 | Response of the detector to commercial arrack liquor (green) containing 40 vol% ethanol and laced with methanol of 0.3 (blue), 0.4 (orange), 0.5 (purple) and 1 vol% (red).

8b) Therefore, what is the detection limit and if not yet reached, what are strategies to further reduce the limit?

In this work, we detected methanol concentrations as low as 0.3 vol% in Arrack (Figure R6) that is sufficient for liquor screening (legal limit in US³⁴ and EU³⁶ is 0.4 vol%). But also lower levels can be detected in lab gas shown in gas mixtures down to 1 ppm (Figure 3). This might be interesting for the production monitoring and quality control of alcoholic beverages (e.g. naturally occurring methanol in wine⁴²).

This information was added on p. 8, par. 2: "... Due to the high signal-to-noise ratio, also lower concentrations of methanol should be detectable, which may be interesting for the production monitoring and quality control of alcoholic beverages (e.g. naturally occurring methanol in wine⁴²). ...” and on p. 14, par. 3: "... Concentrations <0.3 vol% are not relevant for the liquor screening as the legal limit is 0.4 vol% in the US³⁴ and EU³⁶. ...”

Minor issues:

Line 152: Figure 4 (not 3)

We compare here the results of the measurement in methanol/ethanol mixtures (Figure 4a) with the results from the single gas measurements in Figure 3. So, it is correct.

Line 184 normal: Done and thank you for the hint!

Line 190 were: Done.

Response to Referee 3:

This paper is related to development of methanol detector with low cost metal oxide based gas sensor using chemoresistive gas sensor (Pd-doped SnO₂ nanoparticles) to quantify the methanol concentration. Note that this paper used Tenax® Which is particularly useful for the analysis of high boiling compounds such as alcohols, polyethylene glycols, diols, phenols, monoamines and diamines, ethanolamines, aldehydes, ketones and chlorinated aromatics.

1) Tenax will trap other molecules with bigger molecular weight.

This is true and the reason why we utilize Tenax as sorbent. It retains methanol and ethanol just enough to separate them at the optimized device conditions with low sampling flow (25 mL/min) and small column size (4.5 cm). The numerous volatiles with higher molecular weight in breath⁴⁸ or liquor⁴⁷ are retained longer³¹ and do not cause interference as shown from our liquor (Figure 5) and human breath (Figure 6) measurements.

In response to this, we added on p. 10, par. 1: "... Interestingly, in liquor (Figure 5) and human breath (Figure 6), only methanol, ethanol and hydrogen (breath) are clearly detected by the sensor, although both liquor and breath are complex mixtures with more than 100⁴⁷ and 8008 analytes, respectively. This is probably due to the higher molecular weight ..."

2) Therefore, this work need to show the specificity to Methanol over other molecule such as glycol, diols, amine etc. to strongly support the experimental results.

Glycols, diols, amines, etc. are typically retained longer (e.g. ethylene glycol 100 times longer than methanol³¹) and dispersed more in the filter, and thus, do not cause interference during the measurement (see above comment). Even when they elute during subsequent measurements, the sensor is not interfered as the concentrations of these compounds are usually several orders of magnitude lower in breath (e.g. 0.003 ppm trimethylamine⁵⁰ compared to 133 ppm of methanol in case of intoxication¹) or liquor.

In response to this, we added on p. 10, par. 1: "... and different functional groups (e.g. diols or glycols) of most interferants, resulting in longer retention in the filter than methanol (e.g. ethylene glycol 100 times longer than methanol³⁰). Most likely reason, however, is the much lower concentration of most confounders (e.g. 0.003 ppm trimethylamine in breath⁵⁰ compared to >133 ppm of methanol in case of intoxication¹). ..."

3) Another issue needs to address is the drift over time of sensors and absorbent to evident to quantitative performance of this sensor.

Metal-oxide gas sensors are used commercially (e.g. Figaro Sensors) and are quite stable. However, they might show slight drifts during operation. We now evaluated the baseline resistance stability for our sensor over 18 days (Figure R7). The sensor baseline drifts in ambient air with regular testing of liquor headspace by 0.7% per day (dashed line, fitted), in-line with similar flame-made Pt-doped SnO₂ sensors operated under lab conditions for 20 days.²⁷ This could be corrected by an additional processing algorithm.⁴³ The filter should not contribute to this drift since it regenerates completely after each exposure in lab gas mixtures (Figure 2c&d) and liquor (Figure 5) and breath (Figure 6) measurements.

In response, we added Figure R7 as Figure S4 to the Supporting Information and included in the text on p. 8, par. 3: "...The methanol detector features also good stability with a sensor baseline drift of 0.7%

per day during 18 days of testing (Figure S4). Such drifts could be corrected by an additional processing algorithm.⁴³

Figure R7 | Baseline resistance of the methanol detector in room air over 18 days with regular testing of liquor headspace. A linear fit (dashed line) indicates an upward drift of 0.7% per day.

4) This paper will be suitable to more development or application based scheme journals because this paper do not present breakthrough research outputs that will suit for Nature Publication but if the novel material in sensor and absorbent are used , this paper will be able to publish.

We believe that it suits for Nature Communications as we solve an important and long-standing problem. Every year, methanol intoxication kills thousands of victims, most recently >95 people in India as reported this February in the New York Times⁶ and highlighted on p. 1, par. 1.

The present methanol detector represents a first practical solution to rapidly screen for methanol-laced liquor (Figure 5) and methanol intoxication in breath (Figure 6). Our concept and employed materials together with their integration into a hand-held and low-cost device are distinct features of this research report. Previous sensor systems, although sophisticated, failed to demonstrate the required performance under realistic conditions or even in laboratory gas mixtures (please see our response to Referee 1, comment #2). Our detector is compact (Figure 1a), simple-in-use and inexpensive enough to be used by first responders in developing countries where most outbreak occur.

In response, we added to the Summary and Conclusions on p. 11, par. 1: "... and could be used by first responders in developing countries, where most outbreaks occur. ..."

Stab ForschungRämistrasse 101
8092 ZürichDr. Robert Schikowski
HG E 34.3
+41 44 63 25507
robert.schikowski@sl.ethz.chETH Zürich
Prof. Dr. Sotiris E. Pratsinis
Institut für Verfahrenstechnik
ML F 20.2
Sonneggstrasse 3
8092 Zürich

12 June 2019 schi

Your project “Highly Selective Detection of Methanol over Ethanol by a Handheld Gas Sensor”

Dear Prof. Pratsinis,

This is to confirm that based on the ETH Zurich Ethics Commission's statutes (RSETHZ 413), your research project

Highly Selective Detection of Methanol over Ethanol by a Handheld Gas Sensor

is not subject to ethics approval at ETH Zurich.

Kind regards,

Dr. Robert Schikowski
Secretary of the ETH Ethics CommissionProf. Lutz Wingert
Chair of the ETH Ethics Commission

REVIEWERS' COMMENTS:

Reviewer #1 (Remarks to the Author):

The authors responded well to the referees comments and added all the missing information to the manuscript. In my opinion the paper is now scientifically rigorous and the conclusions are based on convincing and reliable data.

The originality is well justified in the author's response, present work is original in the sense of actual practical applicability to breath analysis.

Societal relevance is comparatively high.

The only minor point is a suggestion to replace the word "drunken" throughout with a more scientific term. Perhaps "after ingestion of ethanol".

Patrik Spanel.

Reviewer #2 (Remarks to the Author):

The paper has been significantly improved in the revision process thanks to the competent and convincing responses and changes. From my point of view, the manuscript can be published after minor revision:

I now understood even better after reading the reviewer comments, the rebuttal and the modified manuscript that the term "filter" is critical and requires a brief additional statement in the paper. If the authors insist on calling the material used here "filter" (even if strictly spoken it is not), this should be explained.

In this context, the sentence that is connected to the first mentioning of the filter: "It consists of an absorption filter (Tenax) separating methanol from interferants like ethanol, acetone or hydrogen, as in gas chromatography, and a chemoresistive gas sensor (Pd-doped SnO₂ nanoparticles) to quantify the methanol concentration." is misleading because in gas chromatography, no "filters" are used and the "separation" happens not by filtration but by spreading out components in time. To prevent confusion, this must be changed. This would also be helpful to distinguish this work more clearly from the previous formaldehyde sensor work published by the authors where (as U understood from the rebuttal) a "real" filter was used.

Response to Referees of Manuscript NCOMMS-19-11951A-Z:

Response to Referee 1:

The authors responded well to the referees comments and added all the missing information to the manuscript. In my opinion the paper is now scientifically rigorous and the conclusions are based on convincing and reliable data.

The originality is well justified in the author's response, present work is original in the sense of actual practical applicability to breath analysis.

Societal relevance is comparatively high.

The only minor point is a suggestion to replace the word "drunken" throughout with a more scientific term. Perhaps "after ingestion of ethanol".

Patrik Spanel.

Thank you for your valuable feedback! We agree and changed “drunken” to “intoxicated” on p. 9, par. 2, and in the legends of Figure 6 and Supplementary Figure 3.

Response to Referee 2:

The paper has been significantly improved in the revision process thanks to the competent and convincing responses and changes.

Most welcome and we appreciate your feedback that helped us improve the manuscript.

From my point of view, the manuscript can be published after minor revision:

I now understand even better after reading the reviewer comments, the rebuttal and the modified manuscript that the term "filter" is critical and requires a brief additional statement in the paper. If the authors insist on calling the material used here "filter" (even if strictly spoken it is not), this should be explained. In this context, the sentence that is connected to the first mentioning of the filter: "It consists of an absorption filter (Tenax) separating methanol from interferants like ethanol, acetone or hydrogen, as in gas chromatography, and a chemoresistive gas sensor (Pd-doped SnO₂ nanoparticles) to quantify the methanol concentration." is misleading because in gas chromatography, no "filters" are used and the "separation" happens not by filtration but by spreading out components in time. To prevent confusion, this must be changed. This would also be helpful to distinguish this work more clearly from the previous formaldehyde sensor work published by the authors where (as I understood from the rebuttal) a "real" filter was used.

The term “filter” is commonly used for packed particle beds to separate gas mixtures. Gas mask filters based on such sorption particles are used extensively since the World War I to separate and remove toxic gases (https://en.wikipedia.org/wiki/Gas_mask). Also in combination with gas sensors, filters based on activated carbon particles are applied to remove volatile organic compounds for selective CO detection [Schweizer-Berberich, M. et al. *Sens. Actuators, B* **66**, 34-36 (2000)] while we applied activated alumina particles in a previous study to remove hydrophilic compounds for selective isoprene sensing [van den Broek, J. et al. *ACS Sens.* **3**, 677-683 (2018)]. So, the term is correct.

Nevertheless, in analytical chemistry, there is a distinction between filters (such as membranes) that remove interferants from a gas matrix and separation columns that result in adsorption and release (spread) of all matrix components in time (such as sorption packed beds). Therefore, we changed the term “filter” to “separation column” throughout the paper to avoid any confusion.